# The More Natural the Window, the Healthier the Isolated People—A Pathway Analysis in Xi’an, China, during the COVID-19 Pandemic

**DOI:** 10.3390/ijerph191610165

**Published:** 2022-08-17

**Authors:** Wangqin Bi, Xinyi Jiang, Huijun Li, Yingyi Cheng, Xingxing Jia, Yuheng Mao, Bing Zhao

**Affiliations:** 1The College of Landscape Architecture, Nanjing Forestry University, Nanjing 210037, China; 2The College of Art, Xi’an University of Architecture and Technology, Xi’an 710000, China

**Keywords:** green view, window view, lockdown, anxiety, depression, public health, China

## Abstract

This study explores how windows with a green view might affect the mental health (i.e., depressive/anxiety symptoms) of home-isolated populations. An online survey was conducted among 508 adults isolated under government quarantine policies for COVID-19 emergency pandemic control between 10 and 20 January 2022 in Xi’an, China. Structural equation modeling was employed to identify the pathways from green view through windows to isolated people’s depressive/anxiety symptoms. The relative frequency of plant/water exposure through windows was associated with fewer depressive/anxiety symptoms. Home-isolated people during COVID-19 reported better mental health when they were exposed to more natural settings. These findings could inspire public health authorities to adopt nature-based solutions to mitigate the adverse mental health consequences of isolated populations during the pandemic.

## 1. Introduction

The coronavirus disease (COVID-19) pandemic began in China in late 2019, and has spread worldwide by 2022. Rising numbers of infections and deaths pose a significant challenge to global public health systems. Faced with this public health emergency, governments encouraged or required quarantine policies. Furthermore, some governments adopted a national blockade or partial blockade approach [1]. The first government affected, China, took unprecedented strict measures to achieve zero Wuhan cases after a full 76-day lockdown, followed by a 28-day lockdown in Xi’an to achieve zero new cases. However, COVID-19 is still gradually becoming a global public health crisis with incalculable damage to people’s physical and mental health [2]. Numerous experiments on specific psychiatric disorders during COVID-19 have shown that prolonged social isolation can produce a range of negative emotions, such as irritability, fear, anxiety, and sadness [3,4,5,6,7,8,9], leading to a range of psychological problems such as insomnia, rising depression and anxiety, loneliness, and even more serious suicidal ideation [10,11,12,13]. Moreover, a literature review systematically studied the impact of the COVID-19 pandemic on psychological and psychosocial factors and proposed the “psychological COVID-19 syndrome”, with anxiety, stress, and depression becoming the greatest distress during the epidemic; these psychological problems caused by the COVID-19 pandemic are in urgent need of relief [14]. There is growing evidence suggesting that exposure to green space has positive impacts on human health and well-being, primarily by encouraging physical activity, relieving mental stress, restoring memory and concentration, enhancing a sense of well-being, and improving social cohesion [15,16,17,18,19]. A large number of recent studies have further investigated people’s exposure to nature during COVID-19 and found that green space plays a more important and irreplaceable role in people’s physical and mental health during special times [20]. However, during COVID-19, when strict quarantine policies were implemented, more or less green space access barriers, most pathways to good mental health were not possible [21,22]; therefore, it remains unknown the mechanism and pathway of nature’s promoting mental health [23]. In our present study, we hope to use the natural experimental setting of epidemic isolation to explore whether there are partial mediators to explain the above.

### 1.1. The Influence of Window Views on Physical and Psychological Health

Numerous studies have shown that the view of nature through windows can have a variety of positive impacts on four major groups of people (office workers, students, patients, and prisoners). First, for the working population, a view from the window can better serve the office population, alleviate feelings of isolation, depression, and anxiety, and increase the value of work interest and job satisfaction [24,25]. Second, for student groups, a green scene in the classroom can promote better student performance (attention restoration and stress reduction), resulting in better academic performance [26,27]. For the patient group, a vulnerable group, the more natural the ward window is, the fewer postoperative pain medication doses the patient takes, the stronger the postoperative, the better the recovery, the better the vital signs, the more positive the postoperative mood, and the shorter the postoperative hospital stay [28,29]. Finally, prison research confirms that the naturalness of the window view has a direct impact on the inmate’s physical health, reducing stress symptoms such as digestive disorders and headaches, reducing their visits to the infirmary, and bringing them greater health benefits [30,31].

These four populations share the same characteristics of being in one place for long periods and under supervision, where they are subjected to various stressors and can only relate to the outside environment through windows. This is remarkably similar to home isolation during COVID-19. Together, anxiety and depression increase because of prolonged home isolation. As depression and anxiety disorders are both considered stress-related mental disorders [32], it is urgent to study how the level of green exposure in people’s houses has a positive effect on the population.

We also reviewed articles on window views and mental health during the outbreak and found that time spent on window views was significantly associated with better well-being among people who were isolated at home. Two scholars from Israel and Japan both investigated online the impact of green views through room windows on mental health during the COVID-19 lockdown and found that nature around the home may play a critical role in mitigating the adverse mental health consequences of the pandemic and the measures taken to cope with it during this specific period [23,33]. Another researcher went beyond the green view outside the window and brought the blue space (water) into the study to highlight the potential benefits of blue-green space in ecosystem services for mental health and well-being [34].

Both of these studies demonstrate that the degree of nature in the home window view is positively associated with people’s psychological well-being, suggesting a passive way of accessing nature while there are barriers to green space access. However, the mechanisms by which green landscapes in window views have psychological benefits for populations are unclear.

### 1.2. The Possible Pathways between Window Views and Physical and Psychological Health

These studies have centered on the stress-reduction theory proposed by Roger Ulrich and the attention-recovery theory proposed by Steven Kaplan. In daily life, facing tedious work and boring life will produce directed attention fatigue, and nature can provide a recovery environment. The natural environment can finally achieve attention recovery through four processes, which are “fascination”, “being away”, “extent”, and “compatibility”. A number of articles have demonstrated that being away and fascination can act as mediators of the restorative benefits of window view generation [35,36,37]. In contrast, in stress-reduction theory, when exposed to a nonthreatening natural environment rather than various urban environments, it facilitates psychological and physiological stress recovery, a shift in mood towards positive aspects, and positive physiological activity level changes [38,39].

The theoretical foundations of the current paper are based on both of these theories, and a recent study examining the effects of indoor plants and outdoor greenery on people’s depression and anxiety symptoms during the epidemic further demonstrates the irreplaceable role of being away and the restorative qualities of attention-recovery theory as mediators of mental recovery.

### 1.3. The Hypotheses

In late 2021, when the Chinese government issued an emergency notice to close all neighborhoods (villages) and units in Xi’an from 23 December due to COVID-19. Two weeks after 10 January 2022, we distributed an online questionnaire to the Xi’an population to investigate the associations between green view at home and mental health. We hypothesized the following: (1) the more window views available of plants/water nature during a significant lockdown, the more time and frequency people will look out the window. (2) Window views of plants/water may minimize symptoms of depression and anxiety by reducing feelings of loneliness and improving life satisfaction. (3) This would also largely explain increased feelings of escape and compatibility experiences. The relationship between these hypotheses is presented in the study’s conceptual framework (Figure 1).

## 2. Materials and Methods

### 2.1. Study Area

Xi’an, China, has a population of about 13 million and an area of 10,108 km^2^ (Xi’an Municipal Government 2022). Because of the COVID-19 crisis, the Xi’an City New Crown Pneumonia Epidemic Prevention and Control Command issued a notice stating that Xi’an had closed all neighborhoods (villages) and units in the city from 0:00 on 23 December 2021. People were asked to quarantine their homes to avoid gathering. A family member per household was also required to head out every two days to purchase living supplies. Amusement parks, movie theaters, schools, and other places where people gathered were closed. All of the above measures were mandated by law.

### 2.2. Study Design

In January 2022, we conducted an online questionnaire survey through a market research company among 508 random adults with permanent ID addresses in Xi’an. The online survey was divided into four sections, each with moderately long and brief questions with the expectation that this would enhance the validity of the questionnaire. This questionnaire investigated the relationship between home green view and the mental health of Xi’an people who were isolated at home: (1) green view (degree of window greenness, average daily frequency of looking out the window, and duration of looking out the window); (2) possible impact mediators (restorative quality of windows, life satisfaction, and loneliness); (3) mental health (depression and anxiety); and (4) sociodemographic information (gender, age, education level, and household income). The survey was designed in Chinese and was translated into English within this article. The survey was distributed online between 10–20 January 2022—two weeks after the Xi’an blockade to ensure the authenticity of the questionnaire data.

The market research company conducted screening methods that ensured the data was authentic and valid: (1) comprehensive coverage of the population. Those who could participate in the questionnaire were all active online members of the company’s cooperative sample database, covering all age groups and occupational groups of adults above 18 years of age to ensure the full coverage of the data population; and (2) accurate population screening. First, based on the member tags, the questionnaire was pushed to people with the tag of Xi’an City in it. Second, there was a screening city question in front of the questionnaire to confirm that the answer option is Xi’an City before continuing to answer the questionnaire. Finally, a technical function was used to identify whether the IP address was in Xi’an. Through multiple tests, it is ensured that the members who fill in the questionnaire are the target group who are isolated at home in Xi’an City. (3) Effective quality control of the questionnaire. Each respondent could only answer once, and duplicate IPs could not complete the questionnaire twice. To prevent slack responses, two general knowledge questions were put in the middle and conclusion of the questionnaire, and any questionnaire with slack responses, repeated answers, an answer time of less than 120 s, or clear logical mistakes was labeled an invalid sample.

### 2.3. Mental Health Assessment

Depression and anxiety were self-reported as mental health outcomes by the participants. The four-item Patient Health Questionnaire (PHQ-4) screening scale was used to measure depression and anxiety outcomes [34,40] (Kroenke, Spitzer et al., 2009, Pouso, Borja et al., 2021). This measure consists of two brief screening questionnaires that assess generalized anxiety disorder (GAD-2) and depression (PHQ-2). Kroenke proved in 2007 that the two core items (GAD-2) functioned effectively as an anxiety disorder screening tool. Kroenke tested a 2-item version of the PHQ depression module in 2003 and discovered that the PHQ-2 had good construct and criterion validity, making it a good choice for depression screening [41] (Kroenke, Spitzer et al., 2003). Thus, 0 (not at all), 1 (a few days), 2 (almost half of the days), or 3 (nearly every day) were the PHQ-2 alternatives, with scores ranging from 0 to 6. More severe depression and anxiety were associated with higher scores. The internal consistency of our sample was high (McDonald’s ω = 0.84). Here, an omega (ω) coefficient was employed because Trizano-Hermosilla et al. proposed that α estimates reliability less accurately than ω [42]. The Chinese versions of the PHQ-4 and CAD-2 are both validated and proven scales [43,44].

### 2.4. Green View Assessment

This study’s assessment of green view was based on literature precedents, and all indicators were self-reported. As shown in Table 1, it was measured primarily by the degree of greenness in the window views (plants/water), frequency, and the duration of green viewing through windows [23,45].

### 2.5. Putative Mediator Assessment

We hypothesized three a priori mediators: the restorative quality of the window views, life satisfaction, and loneliness. Among them, the restorative quality of windows contains two dimensions: being away and compatibility. All three of these were hypothesized to be the effects of green views on mental health. The Chinese version of Perceived Restorative Scale was validated and proven [46], while life satisfaction and loneliness scales were translated into Chinese using the back-and-forth translation method.

The Perceived Restorative Scale was used to create the questionnaire questions. For both the being away and compatibility aspects, a single item was employed to decrease the length of the questionnaire and the answer load: “Looking out the window at home gives me a break from my day-to-day routine and I can get away from the things that usually demand my attention” and “Looking out the window at home, I could find ways to enjoy myself in a place like this” with two points (0 = no, 10 = completely) on a scale of 11 [47,48]. The McDonald’s ω for this scale was 0.86.

**Table 1 ijerph-19-10165-t001:** Questionnaire description.

Dimensions	Questionnaire	Literature for Validation
Green view	How much greenspace (plants/water) can you see through the window of your house during lockdown periods?	
	How many times a day do you usually view the greenspace through a window on average during lockdown periods?	Li, H.; Zhang, X.; You, C.; Chen, X.; Cao, Y.; Zhang, G. Can viewing nature through windows improve isolated living? A pathway analysis on Chinese male prisoners during the COVID-19 epidemic. *Front. Psychiatry* **2021**, *12*, 720722. https://doi.org/10.3389/fpsyt.2021.720722. [45]
	How long do you usually view the greenspace through the window on average during lockdown periods?	
Life satisfaction	I am satisfied with my current life during lockdown periods.	Lehberger, M.; Kleih, A.; Sparke, K. Self-reported well-being and the importance of green spaces–A comparison of garden owners and non-garden owners in times of COVID-19. *Landsc. Urban Plan.* **2021**, *212*, 104108. [49]
Loneliness	I often feel lonely during lockdown periods.	Astell-Burt, T.; Hartig, T.; Eckermann, S.; Nieuwenhuijsen, M.; McMunn, A.; Frumkin, H.; Feng, X. More green, less lonely? A longitudinal cohort study. *Int. J. Epidemiol.* **2022**, *51*, 99–110. [50]
Restoration quality of green view through window	Being away: Looking out the window at home gives me a break from my day-to-day routine and I can get away from the things that usually demand my attention.	Dzhambov, A.M.; Lercher, P.; Browning, M.H.E.M.; Stoyanov, D.; Petrova, N.; Novakov, S.; Dimitrova, D.D. Does greenery experienced indoors and outdoors provide an escape and support mental health during the COVID-19 quarantine? *Environ. Res.* **2021**, *196*, 110420. https://doi.org/10.1016/j.envres.2020.110420. [51]
	Compatibility: Looking out the window at home, I could find ways to enjoy myself in a place like this.	Hartig, T.; Korpela, K.; Evans, G.W.; Gärling, T. A measure of restorative quality in environments. *Scand. Hous. Plan. Res.* **1997**, *14*, 175–194. [48]

To explore the life satisfaction of the quarantined groups during the COVID-19 lockdown, we adapted Lehberger’s article and took the item “I am satisfied with my current life during lockdown periods” and answered it on an 11-point scale with two depiction points (0 = not at all satisfied, 10 = fully satisfied) [49].

The measure of loneliness was adapted from a paper published by Thomas et al. in 2021, “I often feel lonely during lockdown periods” and the responses were on a 7-point scale, with two depiction points (1 = strongly disagree, 7 = strongly agree) [50].

### 2.6. Data Analysis

Internal reliability analysis of the questionnaire was performed using McDonald’s ω. Standardization of all continuous variables, including the elimination of univariate outliers and the absence of data outliers. The K-S(L) and S-W tests showed that PHQ-2 and GAD-2 were approximately normally distributed. The data did not adhere to an equal proportional distribution, according to Fisher’s exact test using the chi-square goodness-of-fit.

The correlation between the parameters was investigated using Spearman’s correlation analysis and Welch’s ANOVA. We used a chi-square test of independence to rule out the influence of sociological and demographic information on the mental health of the homebound population and found that the connections between gender, education level, and income on depression and anxiety were not statistically significant. The lack of multicollinearity was demonstrated by tolerance values >0.2, and variance inflation coefficient values of 5.0 [52,53].

More complex models (i.e., multiple mediators) are more suitable for structural equation modeling (SEM) [54]. We chose to show the model more straightforwardly in Amos to facilitate subsequent corrections and adjustments. The restorative quality of the windowed view was decomposed into two separate observed variables: being away and compatibility, with life satisfaction and loneliness also present as observed variables. We developed a latent variable to represent mental health problems, which reflected the observable variables of depression and anxiety. To test the mediation effect, we used the bootstrapping test proposed by Hayes [55]. Bentler (1995) recommended a sample size to free parameter ratio of 5:1. Therefore, the sample size is sufficient. Hu and Bentler established indexes of acceptable model fit in 1999, which were used to assess the goodness-of-fit: nonsignificant X^2^ (*p* > 0.050), X^2^/df < 2.000, root mean square error of approximation (RMSEA) < 0.080, and standardized root mean square residual (SRMSR) ≤ 0.08, Bentler’s comparative fit index (CFI) > 0.900, and the Bentler–Bonett normed fit index (NFI) > 0.900 [56].

Subsequently, we constructed two SEM models based on the two mental health symptoms, depression and anxiety, measured using the PHQ-2 and GAD-2, respectively. Considering the complex relationship between the mediating and dependent variables in this study, we chose to separate the dependent variables, depression, and anxiety, and constructed two separate models, M1 and M2, to prove the hypothesis and better show the pathways of action for how the natural landscape outside the window affects people’s mental health during isolation, including the total, direct, and indirect effect relationships.

Data were processed using SPSS 22.0 and Amos 24.0 (IBM, Armonk, NY, USA) for statistical analysis. Statistical significance was established at *p* < 0.05.

## 3. Results

### 3.1. Descriptive Analyses

A total of 568 questionnaires were collected online, 60 of which were excluded owing to their extreme values, leaving a final sample of 508. These 508 participants were a special sample amid a large blockade in Xi’an, China. Table 2 shows that most of the participants were Chinese females, accounting for 60.6% of the participants; the majority were young and middle-aged, aged 18–25 and 26–35, accounting for 84.8%; 74.4% of the participants had a maximum education level of bachelor’s degree or above, and generally had high educational quality; 82.9% of the participants had a total annual household income of less than 200,000 yuan, and did not have significant poverty. The total annual household income of 82.9% of the participants was concentrated below 200,000, and there was no obvious gap between the rich and the poor.

Regarding green views, 33.30% of participants reported that they could see some nature views (plants/water) from their windows at home. However, the percentage of participants who could see many nature views from their windows at home was also less than 10%. This may be because the survey took place from mid-to-end of January, and the geographical location of Xi’an is north of the Tropic of Cancer—a warm-temperate semi-humid continental monsoon climate where plants are mainly deciduous broad-leaved types. Thus, almost all plants in Xi’an have lost their leaves in January, and the degree of nature outside the participants’ windows is bound to be partially affected. As shown in Table 2, 85.6% of the participants would look out of their windows more than three times a day on average during the COVID-19 lockdown, and 28.50% would look out of their windows over seven times daily on average; 26.20% of the participants would look out of their windows for 3–5 min at a time on average, which is also the time most people choose to look out of their windows. A further 38.8% of the participants said they would look out of their windows for over 5 min at a time. These data indicate that people isolated in their homes during the COVID-19 lockdown were more inclined to seek out and explore the natural environment outside, which was completely different from their houses.

### 3.2. Total Association between Green View, Putative Mediators, and Mental Health

Next, we investigated whether the greenness of the window affected the mental health of the home isolation population. The degree of greenness outside people’s windows was significantly positively correlated with the average frequency and duration of looking out of the window at one time, with correlation coefficients of 0.226 and 0.155 (Table 3), respectively. This indicates that the degree of green outside the window interacts with the time and frequency with which the isolated group looks out of the window. Only window greenness and depression showed a significant negative correlation coefficient of −0.147. However, the hypothesized mediating variables, life satisfaction and loneliness, were both correlated with anxiety, with correlations of −0.301 and 0.392, respectively. The hypothesized mediating variables of life satisfaction, loneliness being away, and restoration showed significant correlations with depression, with correlation coefficients of −0.282, 0.413, −0.119, and −0.137, respectively.

### 3.3. Structural Equation Modeling

To explore the specific mechanisms by which window greenness influences mental health, we created two SEM models, M1 and M2, to investigate how window view affects anxiety and depression, respectively, through the corresponding mediators (Figure 2 and Appendix A). The initial models had poor fit for both models, and we removed nonsignificant confounding paths and added paths indicated by high modification indices to obtain high-fit models. Fit for M1 was X^2^/df = 1.799; NFI = 0.970; IFI = 0.988; CFI = 0.988; RMSEA = 0.040; SRMR = 0.026; fit for M2 was X^2^/df = 1.828; NFI = 0.974; IFI = 0.968; CFI = 0.986; RMSEA = 0.040; SRMR = 0.033.

As shown in Table 4 and Appendix A, significant total effects were observed for loneliness on both depression and anxiety. However, nature views through windows were indirectly associated with depression and anxiety, mainly through feelings of escape, life satisfaction, and loneliness during the lockdown. These results are consistent with our second and third hypotheses.

The particular indirect effects, as proposed by Brown [57], are the most relevant form of impact for measuring mediation in structural equation models. The mediation model was then used to investigate particular indirect effects. As shown in the final M1 model, natural window views affect the mental health of isolated populations, primarily through an indirect effect. The green view outside the window can improve the number of times and duration of looking out of the window of the isolated population, which is consistent with Hypothesis 1. The frequency and duration of looking out of the window directly affect people’s feelings of escape and compatibility. As illustrated in Table 4, viewing the green view outside the window can also improve loneliness during isolation by generating feelings of being away, thus reducing the level of depression, which are the two more significant indirect paths in the model. This shows that loneliness during isolation is a more prevalent negative emotion and that people prefer to escape from real life. Contrastingly, the isolation status quo did not improve mental health by increasing life satisfaction.

Meanwhile, in the final M2, Appendix A shows that looking out the window works mainly through the pathway of reduced loneliness, which then increases life satisfaction and reduces anxiety during isolation. In this model pathway, a more green view of the window leads to greater resilience, higher life satisfaction, and lower anxiety.

## 4. Discussion

### 4.1. General Findings

In the context of rapid urbanization and globalization that may lead to increased epidemics, understanding the benefits of green view on mental health and well-being can help policymakers make more informed decisions for public health [34]. This study has three points. First, having higher green visibility in the living environment of home-isolated populations is more likely to reduce anxiety and depression and improve mental health; increased green visibility in the home implies individuals spend more time looking out the window, promoting the frequency and duration of looking out the window. Second, windows with a green view do not directly improve the mental health of isolated people, but rather reduce anxiety and depressive symptoms by enhancing people’s sense of psychological distance and supporting their focus on restoration. Third, during the lockdown period, loneliness was the most common negative emotion, and looking out of the window with a green view could help them to address these bad emotions. These results suggest that urban nature, as a nature-based solution, can improve public health and suggest a new approach for the public health sector [58].

The first thing to highlight is the current status of individuals residing in Xi’an, China during the quarantine period. For emergency control of the COVID-19 epidemic, the residents of Xi’an were quarantined at home, and their activities were limited to their housing environment. The term “residential environment” refers to a collection of settings that surround living and living areas [59]. Based on Pouso’s criteria for the level of lockdown during the COVID-19 pandemic, we determined the level of lockdown in Xi’an, China to be Level 1; no one was allowed to leave their homes except for a specified number of people who could go out to purchase household goods during a specified time. When confronted with a major crisis, individuals develop a succession of emotional, behavioral, and physiological stress reactions based on their own cognitive appraisal, according to Lazarus and Folkman’s theory of psychological stress [60]. The strict quarantine system imposed by the Chinese government on the entire urban population to block the transmission route of the epidemic, with complete restriction of people’s movement and high or low levels of emotional stress, is a unique natural experimental setting. Our findings are also in line with earlier research demonstrating that persons who live alone experience considerable anxiety and depression symptoms.

Second, this study reveals that viewing nature outside the window can play an important role in avoiding depression and anxiety in people isolated during COVID-19. This supports prior research findings [61]. As mentioned above, regarding the two psychological disorders of anxiety and depression, it is quite understandable that in this study, depression, and anxiety were found to be sex-independent, with middle-aged and young adults being more likely to develop depression and anxiety symptoms. Middle-aged adults confront a significant responsibility of feeding their families, and they are unable to make money to maintain their families as a result of the pandemic, leaving them very exposed to severe life stress. Therefore, the psychological relief work of a major epidemic should focus on young and middle-aged groups, which are vulnerable to psychological problems, and take preventive measures such as material subsidies or psychological counseling in a timely manner [62].

The positive effect of green view on mental health does not occur directly, but we found that it is mediated by a sense of being away and compatibility. This finding is consistent with Kaplan’s theory on attentional recovery [35,37]). Exposure to a greater degree of green increases the sense of psychological distance in isolated populations (away from the residential environment), and immersion in the natural environment can bring high compatibility experiences. In a “confinement”-like life, the natural environment outside the window is very different from the permanent residential environment, and when isolated people spend a more frequent and long time each day on the view outside the window, it means that they desire to be away from the real living environment, from the possible stress of real life, and from the epidemic. In turn, once people become more interested in natural scenery, they are more willing to spend time and energy on the window view and are able to achieve a restoration of concentration and bring about a restorative experience.

Finally, in terms of the pathway of action from the naturalness of the window view to psychological well-being, we also found that life satisfaction and loneliness mediated the connection between the naturalness of the window view and psychological well-being. This finding is also in line with the results of a recent study [45]. Since people’s behavior was extremely restricted during the pandemic lockdown, the only form of entertainment that people could partake in was almost exclusively playing with cell phones, computers, and other electronic devices. There was a significant positive correlation between loneliness and the level of cell phone addiction. Therefore, loneliness has become a common problem in the COVID-19 pandemic [63]. We further screened the pathways from the degree of nature of window view to loneliness and life satisfaction and found that (1) directly reducing loneliness through window nature; (2) directly reducing loneliness by window green view, thus enhancing life satisfaction; (3) the more time and frequency of looking out of the window in a day, the less loneliness and more life satisfaction; (4) gaining concentration by increasing the time and frequency of looking out of the window; and (5) by increasing the time and frequency of looking out of the window, the concentration is restored, which in turn enhances life satisfaction. In conclusion, loneliness and life satisfaction, as mediators of mental health problems, can benefit mental health by enhancing the greenness of looking out of the window or by increasing the frequency and duration of looking out of the window.

### 4.2. Policy Implications

In the wake of the outbreak, which had an incalculable negative impact on global health and the economy, Wuhan and Xi’an, China, both adopted a series of evolving public health measures to contain the outbreak in the short term, demonstrating the importance of appropriate and timely public health interventions in the face of crisis. These include traffic restrictions, cancellation of social gatherings, and home quarantine.

Our study includes important implications for people practicing home containment and for guiding urban plans. We found that people in strict containment generally have negative emotions and mild mental disorders, but it is putting the cart before the horse if we focus on containment to protect people from new crowns at the expense of their normal emotional needs and mental health. Theoretically, health authorities should not neglect the mental health of their citizens while paying attention to their physical health. However, the huge cost is beyond reach, so this study proposes a more economical and feasible way, which encourages isolated people to connect to the natural landscape outside the window as much as possible. This also puts more specific requirements on our urban plans to increase the natural elements of residential areas to ensure a green and blue view in front of every home’s windows.

### 4.3. Strengths and Limitations

Although the sample in this study was selected from a population in Xi’an, China, which has a strict home segregation policy, and excluded some unnecessary variables in the study of the relationship between window view and mental health by limiting the green view to the windows of people’s living environments, which to a certain extent ensures the reliability and authenticity of the study results, there are also corresponding limitations.

First, the cross-sectional nature of our study makes it impossible for us to make causal statements about the observed associations. Further, the data in this study were based on self-reported data, which may lead to bias, including self-translation of the scale, and the understanding of bias due to the different contexts of Chinese and English. In the future, we will conduct more longitudinal studies, such as surveys, to follow the same group of people and obtain more information after the lockdown. Alternatively, experiments could be conducted to obtain more objective health indicators to avoid subjective perception bias, which could affect the study results.

Second, we may have overlooked other visible natural elements in people’s living environments that can also contribute to mental health, such as the possible presence of potted plants in the home [64,65,66]. The positive effects of gardening activities on human health that have been shown in numerous studies, natural landscape wall art, green wallpaper, and wall paint, and natural documentaries viewed on the Internet, among others, which may have an impact on people’s mental health [67]. These are the elements of nature that we have not focused on, and we need to pay more attention to these possible variables in future studies, which may provide a restorative environment for isolated populations and protect their health.

Finally, we did not collect specific characteristics of natural elements, such as the combination of plant types, the ratio of building to the sky, the ratio of plant elements in the window viewport, and the perception of multi-sensory patterns brought by window nature, such as the thermal comfort felt after opening the window, the birds’ songs, and the flowers’ smell. In a future study, we expect to quantify the specific characteristics of natural elements and explore the elements and characteristics of the window landscape that might positively affect health. We hope to apply nature-based solutions to improve mental health and well-being while safeguarding against social isolation after the epidemic is normalized.

## 5. Conclusions

This study examined the effects of indirect green exposure to nature through windows on depression and anxiety during the Great COVID-19 Lockdown. We found that home-isolated people with higher levels of greenness outside of their home windows had better mental health. The visibility of nature outside windows is associated with an increase in the frequency and duration of looking out of windows, as well as with lower loneliness, higher life satisfaction, and less anxiety and depression. Window views work primarily through a sense of distance from daily life and high compatibility experiences. Therefore, the current findings could help isolated people to achieve a more positive emotion and better mental health during COVID-19 by increasing visual contact with nature—a relatively low-cost, efficient, and accessible measure.

## Figures and Tables

**Figure 1 ijerph-19-10165-f001:**
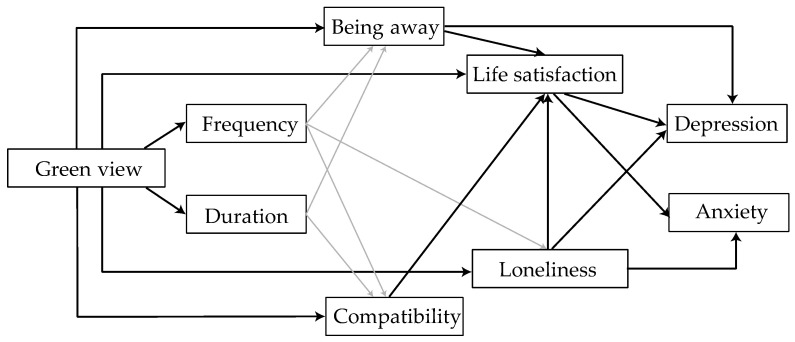
Conceptual framework of green view through a window: Depression and Anxiety. Note: Line widths represent the hypothesized pathway strength, with thicker lines denoting potentially stronger associations.

**Figure 2 ijerph-19-10165-f002:**
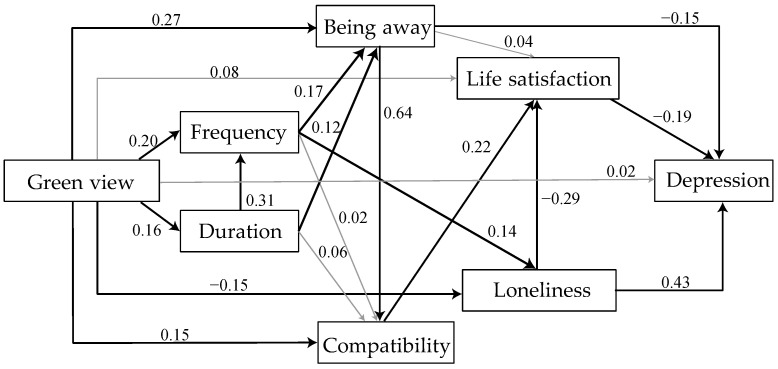
The final SEM model 1 demonstrates standardized effects between variables. Note: The solid black lines indicate *p* < 0.001, and the gray dotted lines indicate *p* > 0.05.

**Table 2 ijerph-19-10165-t002:** Participant characteristics (N = 508).

Characteristics	N	Category	Percentage	Mean (SD)	Range
**Sociodemographics**					
Age (yrs)	241	18–25	47.40%		
	190	26–35	37.40%		
	50	36–45	9.80%		
	23	46–55	4.50%		
	3	56–65	0.60%		
	1	>65	0.20%		
Sex	308	female	61%		
Education	378	university and above	74.40%		
Income (million yuan)	247	10–20	48.60%		
**Mental health**					
Depression (PHQ-2)	508			2.18 (1.27)	0–6
Anxiety (GAD-2)	508			2.16 (1.35)	0–6
**Visibility**					
Green view	45	seldom	8.90%		
	85	few	16.70%		
	159	a bit	31.30%		
	169	some	33.30%		
	50	a lot	9.80%		
Frequency (time)	1	0	0.20%		
	72	1–2	14.20%		
	152	3–5	29.90%		
	138	5–7	27.20%		
	145	>7	28.50%		
Duration (min)	68	<1	13.40%		
	110	1–3	21.70%		
	133	3–5	26.20%		
	106	5–10	20.90%		
	91	>10	17.90%		
**Putative mediators**					
Being away	508			6.75 (2.12)	0–10
Compatibility	508			7.00 (2.10)	0–10
Life satisfaction	508			5.83 (2.18)	0–10
Loneliness	508			4.19 (1.75)	1–7

Note: PHQ-2 = Patient Health Questionnaire-2; GAD-2 = General Anxiety Disorder-2; SD = standard deviation.

**Table 3 ijerph-19-10165-t003:** Spearman’s correlation analysis for the measured parameters.

Variable	1	2	3	4	5	6	7	8	9
1. Green view	1								
2. Frequency	0.226 **	1							
3. Duration	0.155 **	0.333 **	1						
4. Life satisfaction	0.202 **	−0.010	0.054	1					
5. Loneliness	−0.126 **	0.106 *	0.114 *	−0.283 **	1				
6. Being away	0.334 **	0.259 **	0.209 **	0.236 **	0.012	1			
7. Compatibility	0.337 **	0.243 **	0.212 **	0.306 **	0.023	0.652 **	1		
8. Anxiety	−0.047	0.069	0.095 *	−0.301 **	0.392 **	−0.036	−0.083	1	
9. Depression	−0.147 **	−0.043	−0.008	−0.282 **	0.413 **	−0.119 **	−0.137 **	0.693 **	1

Note: Numbers in the cells indicate Spearman’s ρ; * statistically significant at *p* < 0.05 (two-tailed); ** statistically significant at *p* < 0.01 (two-tailed).

**Table 4 ijerph-19-10165-t004:** Effects of green view and mediators on depression in the structural equation model.

Parameters	SE	Estimate	Bias-Corrected Percentile Method		Percentile Method	
			Lower	Upper	Lower	Upper
**Total effects**						
Green view	0.028	−0.070	−0.130	−0.019	−0.130	−0.019
Duration	0.006	−0.006	−0.020	0.005	−0.020	0.005
Frequency	0.013	0.017	−0.007	0.047	−0.009	0.044
Bing away	0.015	−0.044	−0.075	−0.015	−0.077	−0.016
Compatibility	0.004	−0.010	−0.020	−0.004	−0.019	−0.003
Loneliness	0.016	0.138	0.107	0.170	0.107	0.170
Life satisfaction	0.013	−0.043	−0.070	−0.017	−0.068	−0.015
**Indirect effects**						
Green view→Frequency→Being away→Depression	0.002	−0.002	−0.006	−0.004	−0.006	−0.001
Green view→Frequency→Loneliness→Depression	0.002	0.005	0.002	0.012	0.002	0.011
Green view→Frequency→Loneliness→Life satisfaction→Depression	0.001	0.001	0.001	0.002	0.001	0.002
Green view→Duration→Being away→Depression	0.001	−0.001	−0.004	−0.001	−0.004	−0.001
Green view→Being away→Depression	0.008	−0.020	−0.040	−0.004	−0.040	−0.005
Green view→Compatibility→Life satisfaction→Depression	0.001	−0.002	−0.005	−0.001	−0.005	−0.001
Green view→Loneliness→Depression	0.010	−0.030	−0.053	−0.010	−0.052	−0.010
Green view→Loneliness→Life satisfaction→Depression	0.002	−0.004	−0.009	−0.001	−0.007	−0.008

Note: Coefficients are unstandardized linear regression coefficients. SE = bootstrap standard errors.

## Data Availability

The data presented in this study are available on request from the corresponding author.

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
