# Peer review of "The More Natural the Window, the Healthier the Isolated People—A Pathway Analysis in Xi’an, China, during the COVID-19 Pandemic"

_ijerph, 2022, doi:10.3390/ijerph191610165_

Round 1

Reviewer 1 Report

I would like to thank the authors for their work.

This is an interesting paper, which aims to explore how windows with a green view affect the mental health (depressive/anxiety symptoms) of home-isolated populations.

Before the publication, I suggest to change the references as described by the author guidelines: "In the text, reference numbers should be placed in square brackets [ ], and placed before the punctuation; for example [1], [1–3] or [1,3]. References should be described as follows...:. Author 1, A.B.; Author 2, C.D. Title of the article. Abbreviated Journal Name Year, Volume, page range."

There are two result section, I suppose that one of them represent the discussion.

Finally, I suggest to add in the discussion some health policies' perspectives on the basis of the result.

Reviewer 2 Report

The authors investigated through SEM how a green view affects the Mental Health (depressive/anxiety symptoms), Life Satisfaction, and Loneliness of home-isolated populations due to the Covid-19 pandemic. The paper has some merits but also some flaws that make it hard to really understand the magnitude of contribution the authors are providing with their paper. 

In general, I would avoid using terms like "alleviation" of depressive/anxiety symptoms (line 17). The design of your study is only able to identify associations with "fewer" depressive/anxiety symptoms. Although from a theoretical perspective the causation could be plausible, the authors are recommended to use more "smooth" terms to avoid referring to causation.

To enhance your literature background I would suggest adding the following systematic review, especially regarding the anxiety, depression, and other mental health outcomes dimensions. https://doi.org/10.3390/covid2030022

The major concern regarding the influence of window views on physical and psychological health is the fact that both in the introduction and in the results an effect size is never provided. The fact that a significant effect is there, does not imply that the magnitude is meaningful. To help the reader understand the potential importance of window views I would encourage the authors to provide this information while discussing their and other authors' results. Specifically to the authors' results, I would suggest modifying the SEM image by adding explained variance on Depression, Life Satisfaction, and Loneliness. 

Ps: you have two Figure 1. Please correct the numbering. 

Lines 82-94: I found this part too detailed and descriptive. Please try to shorten this part trying to convey the overall meaning and implications of these studies. 

In the Mental health assessment and Putative mediator assessment sub-sections, please provide information about the in-language validation of these scales. In other words, were the scales already validated in the specific Chinese context, or were they translated by the authors? In the latter case, a possible measurement bias should be discussed. 

The results section is double. I mean, the authors called results in both sections 3 and 4. I guess section 4 is the discussion. Please put attention to these elements of the paper. 

In the conclusion section please avoid, once again, causation. Reduces/Enhances is not fine. Please use "associate with higher/lower" or something similar. 

Please note that your citation style is not the one envisaged by MDPI's journals and sooner or later you will have to adapt it. 

Round 2

Reviewer 2 Report

The authors were able to improve their paper based on my suggestions. Some points were addressed better than others but I guess this is the highest level possible of quality the paper could achieve. For this reason, I suggest the paper to be considered for publication.